# The Impact of Emergency Department Arrival Time on Door-to-Balloon Time in Patients with ST-Segment Elevation Myocardial Infarction Receiving Primary Percutaneous Coronary Intervention

**DOI:** 10.3390/jcm12062392

**Published:** 2023-03-20

**Authors:** Yu-Ting Hsiao, Jui-Fu Hung, Shi-Quan Zhang, Ya-Ni Yeh, Ming-Jen Tsai

**Affiliations:** Department of Emergency Medicine, Ditmanson Medical Foundation Chia-Yi Christian Hospital, Chiayi City 600, Taiwan

**Keywords:** ST-segment elevation myocardial infarction, STEMI, door-to-balloon time, temporal difference, nighttime, off-hours

## Abstract

Door-to-balloon (DTB) time significantly affects the prognosis of patients with ST-segment elevation myocardial infarction (STEMI). The effects of temporal differences in emergency department (ED) arrival time on DTB time and on different segments of DTB time remain inconclusive. Therefore, we performed a retrospective study in a tertiary hospital between January 2013 and December 2021 and investigated the relationship between a patient’s arrival time and both their DTB time and different segments of their DTB time. Of 732 STEMI patients, 327 arrived during the daytime (08:01–16:00), 268 during the evening (16:01–24:00), and 137 at night (00:01–08:00). Significantly higher odds of delay in DTB time were observed during the nighttime (adjusted odds ratio (aOR): 2.87; 95% confidence interval (CI): 1.50–5.51, *p* = 0.002) than during the daytime. This delay was mainly attributed to a delay in cardiac catheterization laboratory (cath lab) activation-to-arrival time (aOR: 6.25; 95% CI: 3.75–10.40, *p* < 0.001), particularly during the 00:00–04:00 time range. Age, sex, triage level, and whether patients arrived during the COVID-19 pandemic also had independent effects on different segments of DTB time. Further studies are required to investigate the root causes of delay in DTB time and to develop specific strategies for improvement.

## 1. Introduction

Primary percutaneous coronary intervention (PCI) is the gold-standard therapy for patients with ST-segment elevation myocardial infarction (STEMI) [1,2]. When STEMI patients arrive at the emergency department (ED), timely activation of primary PCI to achieve coronary artery reperfusion is critical [1,2]. Door-to-balloon (DTB) time, a measure of the time required to perform PCI, represents the duration from the patient’s arrival time at the ED to when a balloon is inflated inside the patient’s blocked coronary artery. A longer DTB time has been linked to an increased risk of short- and long-term mortality, major adverse cardiac events, and re-infarction [1,2,3,4,5]. DTB time is a quality indicator used to evaluate the performance of a PCI-capable hospital [1].

National guidelines have recommended that the target time for DTB should be less than 90 min [1,2]. To shorten DTB time, various studies have been conducted that explored the reasons for DTB-time delay [3,6,7,8,9,10,11,12]. Among these, the impact of off-hours or weekends on DTB-time delay represents a contentious issue [12,13,14]. Sorita et al. conducted a systematic review and meta-analysis in 2014, which demonstrated that STEMI patients who arrived during off-hours had longer DTB times than STEMI patients who arrived during on-hours [14]. However, another systematic review by Enezate et al. in 2017 indicated the opposite results [13].

Most studies defined off-hours as the time spent away from work, including the nighttime, weekends, and holiday hours [11,12,13,15]. If only on- and-off hours are used for classification, it may be challenging to determine the period that has the greatest influence on DTB time. Furthermore, the influence of circadian patterns on the incidence of acute myocardial infarction has been previously demonstrated [16]. A three-shift system, operated in most emergency care units, may be better suited for analyzing the impact of a patient’s visit time on treatment goals and determining the target time-period delay.

Taiwan has high medical accessibility and very high universal health insurance coverage. There are 46 accredited advanced emergency-responsibility hospitals across the country that can provide 24/7 emergency cardiac catheterization services [17,18]. These hospitals are required to achieve a DTB time of <90 min for at least 75% of all STEMI patients [1,18]. To the best of our knowledge, there is currently no literature investigating the impact of a patient’s visit time on their DTB time in Taiwan. Furthermore, the time segments of DTB time, which were most affected by the patient’s ED arrival time, are scarcely investigated. The aim of this study is to assess the impact of ED arrival time on DTB time and to identify the key time segments of DTB time that were most significantly influenced by ED arrival time in an advanced emergency-responsibility hospital in Taiwan.

## 2. Materials and Methods

### 2.1. Study Design, Settings, and Data Collection

A retrospective study was conducted between 1 January 2013 and 31 December 2021 in the Ditmanson Medical Foundation Chia-Yi Christian Hospital, a 1000-bed private tertiary referral hospital in the urban city of Taiwan. The hospital is certified as an advanced emergency-responsibility hospital in Taiwan and is capable of performing primary PCI at any time of the day. Their ED receives more than 70,000 patient visits annually.

All patients diagnosed with STEMI in the ED and activated for primary PCI were included in the hospital-based STEMI registry, in which data were prospectively collected for quality improvement purposes. We obtained data for STEMI patients during the study period from the registry and retrospectively collected patient data from electronic medical records. Patients with out-of-hospital cardiac arrest (OHCA) and incomplete data were excluded.

We collected demographic information (age, sex, body weight, body mass index, and medical history), time of ED arrival (season, weekday, or weekend, and whether it occurred during the COVID-19 pandemic years of 2020–2021), mode of delivery (via ambulance or not), ED triage level and vital signs, electrocardiogram (ECG) reports, initial laboratory test results (white blood cell counts and troponin-I levels), echocardiography reports, coronary angiography reports, and hospitalization duration.

For outcome measurement, ECG time, cardiologist consultation time, cardiac catheterization laboratory (cath lab) activation time, the time the patient arrived at the cath lab, and balloon inflation time were obtained. We defined DTB time as the time from ED arrival to balloon inflation, door-to-ECG time as the time from ED arrival to completion of the first ECG, the ECG-to-consultation time as the time from completion of the first ECG to consultation with a cardiologist, the consultation-to-activation time as the time from cardiologist consultation to activation of the cath lab, the activation-to-cath lab-arrival time as the time from activation of the cath lab to the patient’s arrival in the cath lab, and the cath lab-arrival-to-balloon time as the time from the patient’s arrival in the cath lab to balloon inflation (Figure 1) [1,2,3,7,8,9,19]. This study was approved by the Institutional Review Board of the Ditmanson Medical Foundation Chia-Yi Christian Hospital (CYCH-IRB 2022016).

### 2.2. Patient Assignment and Outcome Measurement

To assess the relationship between temporal differences in a patient’s arrival time at the ED with their DTB time as well as different time segments of their DTB time, we divided STEMI patients into three groups based on the time they arrived at the ED (daytime—08:01–16:00; evening—16:01–24:00; and nighttime—00:01–08:00). The primary outcome measure was a DTB time of >90 min. The secondary outcomes were a door-to-ECG time of >10 min, an ECG-to-consultation time of >10 min, a consultation-to-activation time of >10 min, an activation-to-cath lab-arrival time of >30 min, and a cath lab-arrival-to-balloon time of >30 min.

### 2.3. Statistical Analysis

We conducted a preliminary study that included STEMI patients who visited the study hospital between 2013 and 2015 to calculate the required sample size. There were 236 STEMI patients during the 3 years. Among them, 43 patients (18.2%) visited at night. The occurrence rate of a DTB of >90 min was 10.3% during the daytime (baseline) and 23.3% during the nighttime. A two-tailed test size of 5% and power of 95% were applied. We calculated that 688 STEMI patients would be required to reach statistical power. Therefore, we included data from 9 years (2013–2021) to meet this requirement.

We compared and analyzed data from STEMI patients who were admitted to the ED during the day (08:01–16:00), evening (16:01–24:00), and night (00:01–08:00). For continuous variables, a Kruskal–Wallis analysis, expressed as median (25th–75th interquartile range), with a Dunn–Bonferroni post-hoc test or analysis of variance (mean ± standard deviation) and Tukey’s post-hoc test was performed after assessing the data distribution. A chi-squared test was used for categorical variables to evaluate the differences among the groups. To evaluate the net effects of temporal differences (daytime vs. evening vs. nighttime) on a delay in DTB time, logistic regression with a forward-selection Wald test was performed with adjustments for the associated predictors and variables with a *p*-value < 0.1, as derived from the univariate analysis. The adjusted variables included age [10,20,21], sex [10,20,22], visits on the weekends [11,12,13,15], visits during the COVID-19 pandemic [23,24,25,26,27], triage levels [7,28], and the region of infarction [10]. Additionally, logistic regression with adjustments for the aforementioned variables was carried out for secondary outcomes to determine which period within DTB time was primarily affected by ED arrival time. Given the lengthy recruitment period of this study (9 years), we conducted a sensitivity analysis to control for potential confounding factors across different time periods. In addition to adjusting for the aforementioned variables, we performed a multivariate logistic regression with a forward-selection analysis, including the year of recruitment as a covariate. The statistical significance was set at *p* < 0.05. Statistical analysis was performed using JASP Team 2022 (JASP computer software, version 0.16.3).

## 3. Results

### 3.1. Study Population

During the study period, 738 STEMI patients were identified. Five OHCA patients and one patient with incomplete data were excluded. A total of 732 patients were included in this study. A total of 327 patients arrived during the daytime (08:01–16:00), 268 in the evening (16:01–24:00), and 137 at night (00:01–08:00) (Figure 2).

### 3.2. Baseline Characteristics

Comparisons of the baseline characteristics of continuous and categorical variables between the daytime, evening, and nighttime temporal groups are presented in Table 1 and Table 2, respectively. STEMI patients who presented in the daytime were older than those who presented in the evening and nighttime (63.8 ± 12.7 vs. 60.2 ± 13.1 vs. 61.0 ± 13.6 years, respectively, *p* = 0.003; Table 1). The percentage of STEMI patients aged <65 years who arrived during the daytime was significantly lower than those who arrived in the evening and nighttime (51.99 vs. 64.93 vs. 63.50%, respectively, *p* = 0.01; Table 2). STEMI patients who arrived at night had lower body temperatures than those who arrived during the day and evening (35.9 vs. 36.0 vs. 36.0 °C, respectively, *p* = 0.013; Table 1). No significant differences were found among the three temporal groups in other characteristics, such as sex, triage levels, vital signs, laboratory test results, hospitalization duration, ED visit time (seasons, weekdays, or during the COVID-19 pandemic), medical history, and ECG and coronary angiography findings (Table 1 and Table 2).

### 3.3. DTB Time and Time Segments of DTB Time

In terms of DTB time and the time segments of DTB time, STEMI patients who arrived at night had significantly longer DTB times (72 (63–84) min) than those who arrived during the day (57 (47–72) min, *p* < 0.001) and evening (67 (57–76.3) min, *p* = 0.002; Table 1). A significant difference was also observed between the evening and daytime groups (*p* < 0.001). The nighttime group had a significantly longer consultation-to-activation time (median time: 5 min) than the daytime (4 min, *p* = 0.002) and evening (4 min, *p* < 0.001) groups. Moreover, the activation-to-cath lab-arrival time was significantly longer at night (28 min) and in the evening (27 min) than during the day (17 min, both *p* < 0.001).

A significantly higher proportion of patients who arrived at night (16.06%) had a delayed DTB time (>90 min) than those who arrived during the day (6.73%) or evening (10.07%, *p* = 0.008, Table 2). The nighttime group (41.61%) had a significantly higher rate of delayed activation-to-cath lab-arrival time (>30 min) than the evening (30.22%) and daytime (13.15%) groups (*p* < 0.001).

Figure 3 depicts how the frequency of STEMI incidents varies over a 24-h period at 2-h intervals as well as the percentages of delayed time intervals measured. STEMI occurred more frequently during the day, peaking between 08:00–12:00. The major peaks for a delayed DTB time (>90 min) occurred during the hours of 02:00–04:00 (25.0%) and 00:00–02:00 (21.9%), and another small peak occurred during the hours of 18:00–20:00 (14.5%) (Figure 3). A delay in activation-to-cath lab arrival time (>30 min) mainly occurred during the hours of 02:00–04:00 (50.0%), 00:00–02:00 (46.9%), and 18:00–20:00 (46.8%) (Figure 3).

### 3.4. Temporal Differences and the Measured Outcomes

A multivariate analysis was performed to assess the association between ED arrival time in STEMI patients and the occurrence of a delayed DTB time. After adjusting for the variables (*p* < 0.1 from the univariate analysis and the associated factors related to a delayed DTB time as described in the statistical analysis section), STEMI patients who arrived at night had significantly higher odds of having a DTB time of >90 min than those who arrived during the day (adjusted odds ratio (aOR): 2.87; 95% confidence interval (CI): 1.50–5.51, *p* = 0.002; Table 3). Additionally, female STEMI patients also had significantly higher odds of a DTB time of >90 min than male STEMI patients (aOR: 2.44; 95% CI: 1.37–4.32, *p* = 0.002). In order to control for potential confounding factors across different years of recruitment, a sensitivity analysis was performed that replaced the variable “during COVID-19 pandemic” with “year of patient recruitment”. This analysis also yielded similar findings (Appendix A).

Table 4 depicts the assessment of secondary outcomes. STEMI patients who arrived at night (aOR: 6.25; 95% CI: 3.75–10.40, *p* < 0.001) and in the evening (aOR: 3.53; 95% CI: 2.26–5.54, *p* < 0.001) had significantly higher odds of having an activation-to-cath lab-arrival time of >30 min than those who arrived during the day. In contrast, in the other time segments, the odds of delay were not significantly associated with the patient’s arrival time (Table 4). The sensitivity analysis also showed similar findings (Appendix A).

Age (aOR: 1.03; 95% CI: 1.00–1.06, *p* = 0.03) and female sex (aOR: 3.23; 95% CI: 1.63–6.39, *p* < 0.001) were independently associated with door-to-ECG times of >10 min. Patients arriving on the weekend had significantly increased odds of activation-to-cath lab arrival times of >30 min (aOR: 2.11; 95% CI: 1.44–3.11, *p* < 0.001) compared with those arriving on a weekday. Patients who arrived during the COVID-19 pandemic were independently associated with a consultation-to-activation time of >10 min (aOR: 2.06; 95% CI: 1.25–3.41, *p* = 0.005), activation-to-cath lab-arrival time of >30 min (aOR: 4.66; 95% CI: 3.06–7.10, *p* < 0.001), and cath lab-arrival-to-balloon time of >30 min (aOR: 0.52; 95% CI: 0.32–0.86, *p* = 0.011). Patients with triage level 3 also had increased odds of door-to-ECG and ECG-to-consultation times of >10 min compared with patients with triage level 1 (aOR: 3.05; 95% CI: 1.17–7.95, *p* = 0.023; and aOR: 1.69; 95% CI: 1.05–2.72, *p* = 0.031, respectively).

## 4. Discussion

In this study, we assessed whether the ED arrival time of STEMI patients undergoing primary PCI in Taiwan was related to their DTB time and identified the critical time intervals within DTB time that were most significantly influenced by the patient’s arrival time. We found a significantly higher risk of delay in DTB time at night than during the day. The delay between cath lab activation and cath lab arrival primarily caused the delay. The peak time intervals for the delay occurred during the time periods of 00:00–04:00 and 18:00–20:00. Furthermore, age, sex, triage level, and whether the patients arrived during the COVID-19 pandemic also had independent effects on different time intervals within DTB time.

The relationship between off-hours or weekends and a delay in DTB time remains controversial [12,13,14]. In this study, we separated ED visit time and weekends as two variables to analyze their impact on DTB time and the temporal variation at 2-h intervals throughout the day to determine which time period had a higher risk of delayed occurrence. After adjusting for ED visit time and other associated factors, a weekend visit was not significantly associated with a DTB-time delay. However, it was an independent factor in the delay in activation-to-cath lab-arrival time. We also found that not all off-hour periods were associated with a DTB-time delay. The peak periods for delays immediately followed a shift change during non-routine working hours (18:00–20:00 and 00:00–04:00), with the main delay occurring between the time the cardiologist activated the cath lab and the time the patient arrived at the cath lab (activation-to-cath lab arrival). In clinical practice, several factors, including delays due to clinical handovers for ED or cath lab staff, inherent delays for on-call cath lab staff arriving at the hospital, and the limited number of hospital and cath lab staff available during non-routine working hours, may be possible causes. To address this issue, specific hospital strategies, such as concurrently having ED physicians activate the cath lab and consulting a cardiologist, anticipating staff to arrive in the cath lab within 20 min after being paged, and having staff in the ED and cath lab use real-time data feedback, could be implemented [29].

In addition to the temporal effects on DTB time, several factors were independently associated with time segments of DTB time, such as age, female sex, and triage level (Table 4). Atypical presentations were more common in elderly or female patients with acute myocardial infarction [29,30]. As a result, emergency physicians or triage personnel may misjudge and postpone ECG acquisition. Sim et al. investigated the causes of delay in DTB time in Singapore and found that patients in the delayed group were more likely to be older and female than those in the non-delayed group [10]. Dreyer et al. evaluated the sex differences in DTB among STEMI patients in Australia and found that the female sex was independently associated with delayed DTB times after multivariate adjustment [22]. In this study, we further identify that the risk of delayed DTB in elderly and female patients was primarily due to a delay in ECG acquisition (Table 4). This is likely due to the higher likelihood of atypical presentation of acute coronary syndrome in these populations. Similarly, atypical presentations may lead to misclassification as low-acuity triage, resulting in substantial delays in ECG acquisition and interpretation [7,28]. As indicated by our results, a low-acuity triage (level 3) was independently associated with a delay in door-to-ECG and ECG-to-consultation times (Table 4). A high index of suspicion is necessary among ED physicians or triage personnel to minimize delays in door-to-ECG times caused by gender and age disparities.

During the COVID-19 pandemic, the American College of Cardiology, American College of Emergency Physicians, and Society for Cardiovascular Angiography and Interventions still advocated for PCI as the standard of care for STEMI, performed by expert teams wearing personal protective equipment in dedicated cath labs in hospitals with PCI capabilities [30]. The effects of the COVID-19 pandemic on DTB times in STEMI patients have been investigated [23,24,25,26,27,31]. A systematic review and meta-analysis conducted by Chew et al. showed that a significant delay in DTB time was observed during the pandemic, particularly in low-to-middle-income eastern countries [31]. In this study, after multivariate adjustment, patients who arrived during the COVID-19 pandemic were not associated with a delay in DTB time (Table 3). However, there were significantly higher odds of delays in consultation-to-activation time and activation-to-cath lab arrival time during the COVID-19 pandemic (Table 4). Possible reasons for these delays may include stringent infection control in the hospital, the need for personal protective equipment for the consulted cardiologist or cath lab staff, and the requirement for COVID-19 testing before transferring patients to the cath lab [31].

This study has some limitations. First, it was a single-center study with a small number of patients included, and, therefore, our findings may not be generalizable to other hospitals in Taiwan or in other regions. Second, selection bias and unmeasurable confounding factors were inevitable because this was a retrospective study. Third, this study could only determine the period during which delays mostly occurred, yet we were unable to conclusively determine the reason for this delay. Nevertheless, identifying the period with the highest risk of delay can help improve quality control further by determining the root causes and proposing an improvement plan. Additionally, the study spanned a nine-year period, during which there were potential confounding factors, such as changes in guidelines, hospital polices, and staff, all of which may have influenced the results. However, our sensitivity analysis, which controlled for the years of patient recruitment, showed consistent results.

## 5. Conclusions

The current study showed that DTB time was significantly influenced by the time of a patient’s arrival. Patients who arrived at the ED at night were more likely to have a delayed DTB time than those who arrived during the day. The time delay between the activation at and arrival to the cath lab was the primary cause of this delay. Further studies are required to investigate the root causes and develop improvement strategies.

## Figures and Tables

**Figure 1 jcm-12-02392-f001:**
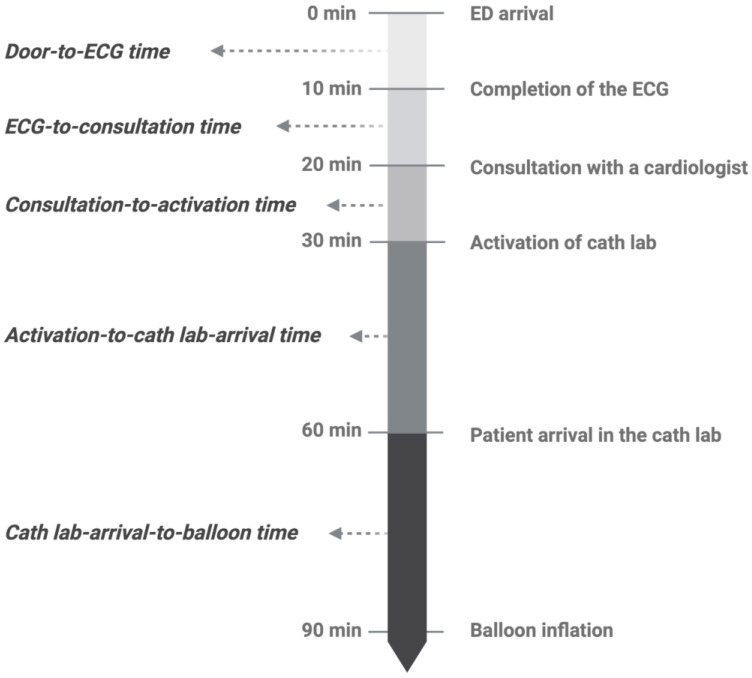
The definition of door-to-balloon time and time segments of door-to-balloon time. ECG—electrocardiogram; ED—emergency department.

**Figure 2 jcm-12-02392-f002:**
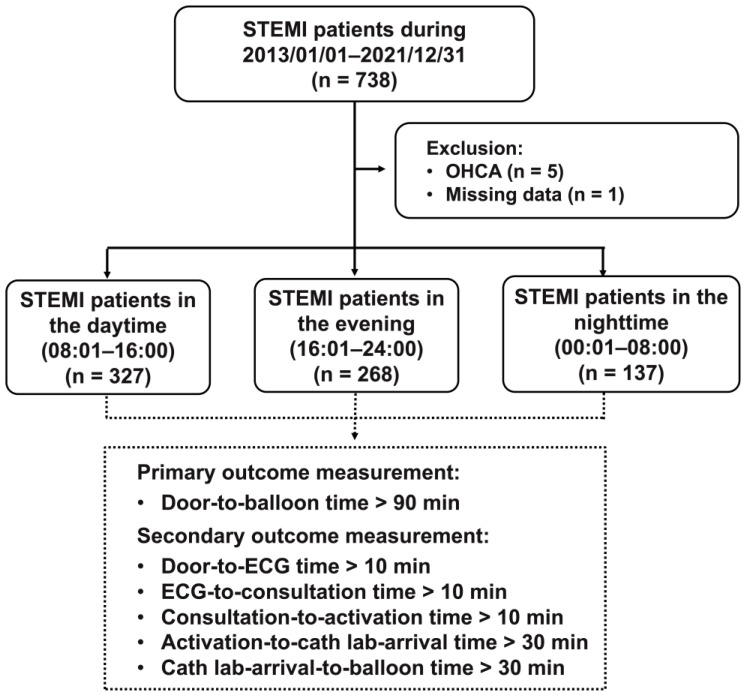
Flowchart of the patients included in the study. ECG—electrocardiogram; OHCA—out-of-hospital cardiac arrest; STEMI—ST-segment elevation myocardial infarction.

**Figure 3 jcm-12-02392-f003:**
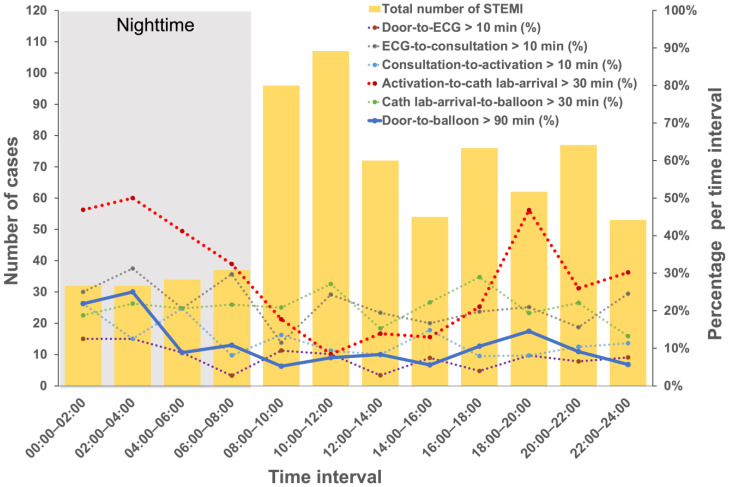
Temporal variations in the number of STEMI incidents and the percentage of measured outcomes at 2-h intervals throughout the day. ECG—electrocardiogram; STEMI—ST-segment elevation myocardial infarction.

**Table 1 jcm-12-02392-t001:** Baseline characteristics and measured time intervals within door-to-balloon time (continuous variables) among three temporal groups.

	Time of ED Visit									
		Daytime 08:01–16:00 (*n* = 327)			Evening 16:01–24:00 (*n* = 268)			Nighttime 00:01–08:00 (*n* = 137)		*p*-Value
**Demographics**										
**Age (year)**	63.8	(12.7)		60.2*	(13.1)		61.0	(13.6)		0.003
**Body weight (kg)**	67.7	(60.0–76.2)	(*n* = 325)	68.2	(60.0–77.0)	(*n* = 265)	70.2	(59.9–80.0)	(*n* = 134)	0.284
**BMI**	25.1	(3.5)	(*n* = 325)	25.5	(4.2)	(*n* = 265)	25.7	(4.3)	(*n* = 134)	0.284
**Triage vital signs**										
**BT (°C)**	36.0	(35.5–36.4)	(*n* = 326)	36.0	(35.5–36.5)		35.9 *^,#^	(35.2–36.2)		0.013
**HR (bpm)**	72.0	(60.0–87.0)		76.0	(65.0–89.0)	(*n* = 265)	75.0	(61.0–89.0)		0.074
**SBP (mmHg)**	133.0	(32.4)	(*n* = 326)	136.8	(32.7)	(*n* = 264)	132.3	(33.3)		0.282
**DBP (mmHg)**	84.3	(20.3)	(*n* = 326)	86.8	(21.7)	(*n* = 264)	84.6	(19.4)		0.336
**SpO_2_ (%)**	97.0	(95.0–98.0)	(*n* = 177)	97.0	(96.0–99.0)	(*n* = 125)	97.0	(95.0–98.0)	(*n* = 79)	0.885
**Laboratory test results**										
**WBC count (×1000/uL)**	10.4	(8.3–13.0)		10.5	(8.7–12.2)		10.4	(8.4–13.2)		0.989
**Troponin-I level (ng/mL)**	0.2	(0.0–3.8)	(*n* = 325)	0.1	(0.0–2.2)		0.2	(0.0–2.9)		0.533
**EF for cardiac echo (%)**	57.0	(48.0–65.0)	(*n* = 313)	56.0	(47.1–65.0)	(*n* = 261)	57.4	(46.9–63.3)	(*n* = 135)	0.824
**Duration of hospitalization (days)**	5.0	(4.0–6.0)		5.0	(4.0–6.0)		4.0	(3.0–6.0)		0.256
**Time interval (min)**										
**Door-to-ECG time**	4.0	(3.0–6.0)		5.0	(3.0–6.0)		5.0	(3.0–6.0)		0.872
**ECG-to-consultation time**	5.0	(2.0–9.0)		5.0	(3.0–8.0)		5.0	(3.0–12.0)		0.161
**Consultation-to-activation time**	4.0	(2.0–7.0)		4.0	(2.0–6.0)		5.0 *^,#^	(3.0–9.0)		<0.001
**Activation-to-cath lab-arrival time**	17.0	(13.0–25.0)		27.0 *	(19.8–33.0)		28.0 *	(22.0–35.0)		<0.001
**Cath lab-arrival-to-balloon time**	21.0	(15.0–29.0)		23.0	(17.0–28.3)		23.0	(18.0–28.0)		0.063
**Door-to-balloon time**	57.0	(47.0–72.0)		67.0 *	(57.0–76.3)		72.0 *^,#^	(63.0–84.0)		<0.001

Data are expressed as mean (standard deviation) or median (25th–75th interquartile range). * *p* < 0.05 compared with daytime; # *p* < 0.05 compared with evening. BMI—body mass index; BT—body temperature; DBP—diastolic blood pressure; ECG—electrocardiogram; ED—emergency department; EF—ejection fraction; HR—heart rate; SpO_2_—pulse oximeter oxygen saturation; SBP—systolic blood pressure; WBC—white blood cell.

**Table 2 jcm-12-02392-t002:** Baseline characteristics and measured outcomes (categorical variables) among three temporal groups.

	Time of ED Visit						
	Daytime 08:01–16:00 (*n* = 327)		Evening 16:01–24:00 (*n* = 268)		Nighttime 00:01–08:00 (*n* = 137)		*p*-Value
	*n*	(%)	*n*	(%)	*n*	(%)	
**Age group**							0.010
**<65 years**	170	(51.99)	174	(64.93)	87	(63.50)	
**65–75 years**	92	(28.13)	52	(19.40)	24	(17.52)	
**≥75 years**	65	(19.88)	42	(15.67)	26	(18.98)	
**Female sex**	56	(17.13)	45	(16.79)	18	(13.14)	0.544
**Season**							0.256
**Spring**	77	(23.55)	61	(22.76)	40	(29.20)	
**Summer**	76	(23.24)	68	(25.37)	24	(17.52)	
**Fall**	86	(26.30)	63	(23.51)	43	(31.39)	
**Winter**	88	(26.91)	76	(28.36)	30	(21.90)	
**ED visit time**							0.876
**Weekday**	228	(69.72)	192	(71.64)	97	(70.80)	
**Weekend**	99	(30.28)	76	(28.36)	40	(29.20)	
**During COVID-19 pandemic (2020–2021)**	76	(23.24)	51	(19.03)	24	(17.52)	0.274
**Ambulance-transported patient**	88	(26.91)	67	(25.00)	33	(24.09)	0.776
**Triage level**							0.094
**1**	28	(8.56)	28	(10.45)	23	(16.79)	
**2**	268	(81.96)	221	(82.46)	102	(74.45)	
**3**	31	(9.48)	19	(7.09)	12	(8.76)	
**Medical history**							
**DM**	131	(40.06)	102	(38.06)	55	(40.15)	0.864
**HTN**	213	(65.14)	171	(63.81)	87	(63.50)	0.920
**Hyperlipidemia**	197	(60.24)	168	(62.69)	77	(56.20)	0.450
**CVA**	27	(8.26)	22	(8.21)	6	(4.38)	0.304
**CKD**	31	(9.48)	14	(5.22)	12	(8.76)	0.140
**ESRD**	9	(2.75)	8	(2.99)	1	(0.73)	0.344
**CAD**	51	(15.60)	31	(11.57)	19	(13.87)	0.366
**COPD**	13	(3.98)	6	(2.24)	3	(2.19)	0.385
**PAOD**	4	(1.22)	2	(0.75)	3	(2.21)	0.454
**Smoking**	192	(58.72)	167	(62.31)	80	(58.39)	0.616
**ECG report**							
**Anterior STEMI**	151	(46.18)	124	(46.27)	61	(44.53)	0.938
**Inferior STEMI**	160	(48.93)	138	(51.69)	67	(49.26)	0.786
**Lateral STEMI**	14	(4.28)	11	(4.12)	0	(0.00)	0.051
**Posterior STEMI**	3	(0.92)	7	(2.62)	2	(1.47)	0.263
**LBBB**	1	(0.31)	1	(0.37)	2	(1.46)	0.273
**AV block**	28	(8.56)	18	(6.72)	14	(10.22)	0.453
**Findings of coronary angiography**							
**1-vessel disease**	88	(27.00)	83	(30.97)	49	(35.77)	0.527
**2-vessel disease**	123	(37.73)	98	(36.57)	41	(29.93)	
**3-vessel disease**	114	(34.97)	86	(32.09)	47	(34.31)	
**LM occlusion**	12	(3.67)	16	(5.97)	7	(5.11)	0.416
**LAD occlusion (≥50%)**	262	(80.12)	215	80.22)	109	(79.56)	0.987
**LCX occlusion (** **≥50%)**	170	(51.99)	125	(46.64)	68	(49.64)	0.431
**RCA occlusion (** **≥50%)**	226	(69.11)	187	(69.78)	88	(64.23)	0.494
**Time interval**							
**Door-to-ECG > 10 min**	24	(7.34)	17	(6.34)	12	(8.76)	0.672
**ECG-to-consultation > 10 min**	59	(18.04)	53	(19.78)	37	(27.01)	0.087
**Consultation-to-activation > 10 min**	37	(11.31)	25	(9.33)	21	(15.33)	0.197
**Activation-to-cath lab-arrival > 30 min**	43	(13.15)	81	(30.22)	57	(41.61)	<0.001
**Cath lab-arrival-to-balloon time > 30 min**	72	(22.02)	58	(21.64)	28	(20.44)	0.931
**Door-to-balloon time > 90 min**	22	(6.73)	27	(10.07)	22	(16.06)	0.008

AV—atrioventricular; CAD—coronary artery disease; CKD—chronic kidney disease; COPD—chronic obstructive pulmonary disease; CVA—cerebrovascular accident; DM—diabetes mellitus; ED—emergency department; ECG—electrocardiogram; ESRD—end-stage renal disease; HTN—hypertension; LAD—left anterior descending artery; LBBB—left bundle branch block; LCX—left circumflex artery; LM—left main artery; PAOD—peripheral arterial occlusive disease; RCA—right coronary artery; STEMI—ST-segment elevation myocardial infarction.

**Table 3 jcm-12-02392-t003:** Predictors associated with door-to-balloon times of >90 min in STEMI patients who arrived to the ED.

Parameters	OR	95% CI	*p*-Value	aOR	95% CI	*p*-Value
**Age (per year)**	1.02	(1.00–1.04)	0.019		–	
**Female sex**	2.41	(1.39–4.19)	0.002	2.44	(1.37–4.32)	0.002
**Weekend**	0.94	(0.54–1.61)	0.815		–	
**During COVID-19 pandemic**	0.60	(0.30–1.21)	0.155		–	
**Triage level**						
**1**		Reference			Reference	
**2**	0.58	(0.28–1.21)	0.146	0.67	(0.32–1.40)	0.283
**3**	2.20	(0.91–5.32)	0.079	2.42	(0.97–6.00)	0.058
**Lateral wall STEMI**	0.80	(0.19–3.47)	0.767		–	
**ED visit time**						
**08:01–16:00**		Reference			Reference	
**16:01–24:00**	1.55	(0.86–2.80)	0.142	1.66	(0.91–3.03)	0.097
**00:01–08:00**	2.65	(1.41–4.97)	0.002	2.87	(1.50–5.51)	0.002

aOR—adjusted odds ratio; CI—confidence interval; ED—emergency department; OR—odds ratio; STEMI—ST-segment elevation myocardial infarction.

**Table 4 jcm-12-02392-t004:** Predictors associated with different time segments of door-to-balloon time in STEMI patients who arrived to the ED.

	Door-to-ECG > 10 min			ECG-to-Consultation > 10 min			Consultation-to-Activation > 10 min			Activation-to-Cath Lab-Arrival > 30 min			Cath Lab-Arrival-to-Balloon Time > 30 min		
Parameters	aOR	95% CI	*p*-Value	aOR	95% CI	*p*-Value	aOR	95% CI	*p*-Value	aOR	95% CI	*p*-Value	aOR	95% CI	*p*-Value
**Age (per year)**	1.03	(1.00–1.06)	0.030		–			–			–			–	
**Female sex**	3.23	(1.63–6.39)	<0.001		–			–			–			–	
**Weekend**		–			–			–		2.11	(1.44–3.11)	<0.001		–	
**During COVID-19 pandemic**		–			–		2.06	(1.25–3.41)	0.005	4.66	(3.06–7.10)	<0.001	0.52	(0.32–0.86)	0.011
**Triage level**															
**1**		Reference			Reference										
**2**	0.44	(0.19–1.00)	0.049	1.14	(0.76–1.73)	0.530		–			–			–	
**3**	3.05	(1.17–7.95)	0.023	1.69	(1.05–2.72)	0.031		–			–			–	
**Lateral wall STEMI**		–			–			–			–			–	
**ED visit time**															
**08:01–16:00**		Reference			Reference			Reference			Reference			Reference	
**16:01–24:00**	1.02	(0.51–2.03)	0.956	0.85	(0.48–1.50)	0.567	0.84	(0.49–1.43)	0.516	3.53	(2.26–5.54)	<0.001	0.94	(0.63–1.39)	0.751
**00:01–08:00**	1.38	(0.63–3.01)	0.425	1.60	(0.75–3.41)	0.229	1.51	(0.84–2.70)	0.168	6.25	(3.75–10.40)	<0.001	0.89	(0.54–1.46)	0.638

aOR—adjusted odds ratio; CI—confidence interval; ED—emergency department; OR—odds ratio; STEMI—ST-segment elevation myocardial infarction.

## Data Availability

The original contributions presented in the study are included in the article, and Appendix A and further inquiries can be directed to the corresponding author(s).

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
