# Peer review of "The Impact of Emergency Department Arrival Time on Door-to-Balloon Time in Patients with ST-Segment Elevation Myocardial Infarction Receiving Primary Percutaneous Coronary Intervention"

_jcm, 2023, doi:10.3390/jcm12062392_

Round 1
Reviewer 1 Report
Comments to the Author:
Although the authors describe in detail the door-to-balloon time according to time periods, there are more studies with such results and they include Meta-analyses. This study lacked innovation and it was a single center study with a small number of patients included.
Also the time span of this study is large and changes in institutional processes, changes in transport, how many surgeons, the distance from home to the unit, etc. during this time period are all factors that the authors need to consider. The authors should refer to previous studies and analyze the reasons and mechanisms for the different results from them.
Author Response
RESPONSE TO REVIEWERS
We are very grateful to the reviewers for their constructive comments and suggestions, which have helped improve our manuscript. We have carefully revised the manuscript and provided point-to-point responses to their comments.
Reviewer 1:
Comments to the Author:
Although the authors describe in detail the door-to-balloon time according to time periods, there are more studies with such results and they include Meta-analyses. This study lacked innovation and it was a single center study with a small number of patients included.
Also the time span of this study is large and changes in institutional processes, changes in transport, how many surgeons, the distance from home to the unit, etc. during this time period are all factors that the authors need to consider. The authors should refer to previous studies and analyze the reasons and mechanisms for the different results from them.
Response: Thank you for your valuable feedback. We acknowledge that this study has limitations, as pointed out by the reviewer. However, healthcare resources and practices vary across different regions and countries, and therefore, it is important to study this issue independently in each region. While there are published meta-analyses on the topic, there is still no consensus about weekend or off-hour effect on DTB time.
To the best of our knowledge, there are currently no studies in Taiwan that investigate the temporal difference on DTB time, nor have previous studies examined the impact of nighttime effect on individual time intervals within the DTB time period. Therefore, we believe that our findings can still provide some information for root cause investigation and improvement strategies.
Regarding the small sample size, we have performed a sample size calculation to ensure adequate statistical power. However, we understand that large studies with multicenter designs are needed to confirm our findings and to draw more generalizable conclusions, as we stated in the limitation in the revised manuscript. (Line: 325-326)
Regarding the lengthy time span of this study, we were unable to control for all potential confounding factors that may have existed during the years of recruitment, such as changes in institutional processes or transportation, etc. However, during the study period, our hospital has been certified as an advanced emergency-responsibility hospital in Taiwan and was required to perform regular quality management. The locations of the emergency department and cardiac catheterization laboratory did not change during the study period. To address concerns raised by the reviewer and minimize potential confounding factors across different years of recruitment, we conducted a sensitivity analysis to adjust for “year of patient recruitment” as a covariate. The analysis yielded consistent results and is presented in supplementary table 1 in the revised manuscript. In addition, we have added reviewer’s concern to the limitation section (line 332-335).
In this study, we believe that in addition to the finding of temporal effect on DTB time, our results have identified other factors that were independently associated with specific time segments within DTB time, such as age, female sex and triage level. Although previous studies have also found associations between these factors and DTB time, our study further identified the specific time-segment within the DTB time. As discussed in the third paragraph of the discussion section, age, female sex, and low-acuity triage influenced the DTB time primarily due to a delay in ECG acquisition. We have reinforced this finding in the revised manuscript (line: 302-309).
Hoping that our response has addressed your concerns, and thank you again for your valuable suggestion.
Reviewer 2 Report
This is a single center retrospective study assessing the relationship between time of arrival at the emergency room and door-to-balloon times (DBT) in STEMI patients. The study found that delayed DBT (>90 minutes) was more likely to occur at nighttime compared to daytime, and was most commonly due to delay in time from activation of the Cath lab to patient arrival at the Cath lab.
Independent predictors of delayed DTB included age, female gender and ED night time arrival.
The study methods are appropriate, and the results are well laid out and easy to understand. By breaking down the DTB into segments, the causes of delays are elucidated. The conclusions are supported by the results.
Specific comments:
1. It is very interesting that the delay in females is seen only at the door to ECG segment. This is most likely explained by the higher likelihood of atypical presentation of acute coronary syndrome in women. A high index of suspicion is necessary among ED physicians or triage personnel to reduce the gender disparity
2. Do the authors plan to look at short and longterm outcomes associated with overall delay and delay in segments? Delay in which segment is associated with worse outcomes? I assume that is beyond the scope of this manuscript.
Author Response
RESPONSE TO REVIEWERS
We are very grateful to the reviewers for their constructive comments and suggestions, which have helped improve our manuscript. We have carefully revised the manuscript and provided point-to-point responses to their comments.
Reviewer 2:
Comments to the Author:
This is a single center retrospective study assessing the relationship between time of arrival at the emergency room and door-to-balloon times (DBT) in STEMI patients. The study found that delayed DBT (>90 minutes) was more likely to occur at nighttime compared to daytime and was most commonly due to delay in time from activation of the Cath lab to patient arrival at the Cath lab.
Independent predictors of delayed DTB included age, female gender and ED night time arrival.
The study methods are appropriate, and the results are well laid out and easy to understand. By breaking down the DTB into segments, the causes of delays are elucidated. The conclusions are supported by the results.
Specific comments:
- It is very interesting that the delay in females is seen only at the door to ECG segment. This is most likely explained by the higher likelihood of atypical presentation of acute coronary syndrome in women. A high index of suspicion is necessary among ED physicians or triage personnel to reduce the gender disparity.
Response: Thank you for your kind words and suggestion. We very agree with your comments. We have added this opinion into our discussion to reinforce our findings. Please see the revised manuscript (line: 302-309).
- Do the authors plan to look at short and longterm outcomes associated with overall delay and delay in segments? Delay in which segment is associated with worse outcomes? I assume that is beyond the scope of this manuscript.
Response: Thank you for your valuable feedback. We appreciate your suggestion to explore the short and long-term outcomes associated with overall delay and delay in segments, as well as to investigate which specific delay segment is associated with worse outcomes. While this was not originally within the scope of our manuscript, we agree that it is a relevant and important area of investigation. We will take your suggestion into consideration and explore the possibility of conducting further research on this topic in the future. Thank you again for your insightful feedback.
Round 2
Reviewer 1 Report
Accept in present form.